# Transcriptional Control of Regulatory T Cells in Cancer: Toward Therapeutic Targeting?

**DOI:** 10.3390/cancers12113194

**Published:** 2020-10-30

**Authors:** Pierre Stéphan, Raphaëlle Lautraite, Allison Voisin, Yenkel Grinberg-Bleyer

**Affiliations:** Cancer Research Center of Lyon, UMR INSERM 1052, CNRS 5286, Université Claude Bernard Lyon 1, Centre Léon Bérard, 69008 Lyon, France; pierre.stephan@lyon.unicancer.fr (P.S.); raphaelle.lautraite@inserm.fr (R.L.); allison.voisin@inserm.fr (A.V.)

**Keywords:** regulatory T cells, cancer, transcription, immunotherapy

## Abstract

**Simple Summary:**

Cancer outcomes are often indexed to the quality of the immune response to the tumor. Among immune cells, Foxp3^+^ regulatory T cells (Treg cells) are potent inhibitors of cancer immunity, and their presence within solid tumors is generally associated with a poor prognosis. Thus, understanding how Treg cell identity is controlled, is of utmost importance for the development of novel anti-cancer therapies. In this review, we summarize the current knowledge on the different intracellular pathways involved in the programming of Treg cell homeostasis and functions in cancer. We also highlight the therapeutic approaches aiming at targeting these regulators to enhance anti-tumor immunity.

**Abstract:**

Extensive research in the past decades has highlighted the tight link between immunity and cancer, leading to the development of immunotherapies that have revolutionized cancer care. However, only a fraction of patients display durable responses to these treatments, and a deeper understanding of the cellular and mechanisms orchestrating immune responses to tumors is mandatory for the discovery of novel therapeutic targets. Among the most scrutinized immune cells, Forkhead Box Protein P3 (Foxp3)^+^ Regulatory T cells (Treg cells) are central inhibitors of protective anti-tumor immunity. These tumor-promoting functions render Treg cells attractive immunotherapy targets, and multiple strategies are being developed to inhibit their recruitment, survival, and function in the tumor microenvironment. In this context, it is critical to decipher the complex and multi-layered molecular mechanisms that shape and stabilize the Treg cell transcriptome. Here, we provide a global view of the transcription factors, and their upstream signaling pathways, involved in the programming of Treg cell homeostasis and functions in cancer. We also evaluate the feasibility and safety of novel therapeutic approaches aiming at targeting specific transcriptional regulators.

## 1. Introduction

Forkhead Box Protein P3 (Foxp3)^+^ regulatory T (Treg) cells compose 5–20% of the total CD4^+^ T cell pool. Their primary and most described function is to maintain immune tolerance and prevent autoimmunity at all time [1]. This is illustrated by the systemic autoimmune syndrome observed in Scurfy mice and Immunodysregulation Polyendocrinopathy Enteropathy X-linked (IPEX) patients who carry mutations in *Foxp3* and after the ablation of Treg cells in young and adult mice [2,3,4,5]. In addition, through their multiple mechanisms of suppression, Treg cells are involved in the inhibition of a wide variety of immune responses, ranging from infection to cancer immunity [6]. Studies conducted in preclinical murine models have established the deleterious function of Treg cells in cancer. Indeed, genetic and antibody-mediated depletion of Treg cells enhances tumor immunity and reduces tumor burden in many settings [7,8]. These conclusions have been largely confirmed in cancer patients, where the accumulation of Treg cells in the blood and tumor tissues is generally indicative of poor prognosis, though several exceptions, such as colorectal cancer, have been identified [9]. Because of this deleterious facet, the development of therapies aiming at modulating Treg recruitment, accumulation, and function in the tumor microenvironment is an area of extensive investigation in the field of cancer immunotherapy. As a prominent example, anti-Cytotoxic T-Lymphocyte-Associated Protein 4 (CTLA-4) antibodies, the first approved checkpoint-blockade therapy for cancer, were shown to exert their beneficial effects in cancer by decreasing Treg cells in mouse models [10], though the relevance of this mechanism in patients is still under debate [11,12]. The effect of Programmed Death-1 (PD-1) blockade on Treg cells and its contribution to therapeutic efficacy is also under scrutiny (reviewed in [13]). Interestingly, it was suggested that PD-1 inhibition on Treg cells may contribute to the hyperprogressive disease observed in a number of patients with gastric cancer [14]. Together, this demonstrates the central role of Treg cells in cancer immunotherapy. Cutting-edge technologies now provide scientists with the ability to comprehend the complexity of Treg cell populations and their molecular regulation to highlight additional therapeutic targets.

## 2. An Overview of Treg Cell Subsets and Their Transcriptional Regulation

The existence of different flavors of Treg cells underlies their large panel of functions. First, Treg cells can either develop in the thymus (tTreg) or differentiate in peripheral lymphoid tissues from naïve conventional (Tconv) cells (pTreg cells and their in vitro relatives, iTreg). To date, whether these two populations rely on shared or distinct transcription factor activity remains unclear. The proper development of Treg cells relies on a large number of transcriptional and epigenetic regulators, either for their survival or for the expression of Foxp3 or its stabilization. These mechanisms have been largely deciphered elsewhere [15,16], and we will therefore focus our review on the transcriptional regulation of mature Foxp3^+^ Treg cells.

Treg cell subsets can also be defined based on their activation status. Whereas naïve-like Resting cells (rTreg) are primarily found in lymphoid tissues, engagement of the T-Cell Receptor (TCR) and its co-stimulation partner CD28, as well as members of the Tumor Necrosis Factor Receptor SuperFamily (TNFRSFs), drives the maturation of rTreg cells to a highly immunosuppressive Activated subset (aTreg cells, also known as effector eTreg cells) [17]. aTreg cells migrate to non-lymphoid tissues, where they maintain tissue homeostasis and potently suppress ongoing immune responses. In particular, aTreg cells are highly abundant in the tumor microenvironment and express a large panel of immune checkpoints (i.e., inhibitory and stimulatory surface receptors), making their regulation an important aspect in the development of immune checkpoint-blockade therapies [18,19] (Figure 1).

This immense regulatory power is under tight transcriptional control. Foxp3 is the master transcription factor of Treg cells, as it provides them with their suppressive function and largely establishes their transcriptomic pattern [20,21]. Moreover, Foxp3 can interact with a myriad other transcriptional regulators, thereby enabling potent repression or activation of gene expression [22,23]. Downstream of the TCR/CD28/TNFRSF signaling cascades, a number of transcription factors are essential for the specification of the aTreg subset. Briefly, Interferon Regulatory Factor 4 (IRF4), Myb, and Nuclear Factor Kappa-light-chain-enhancer of activated B cells (NF-κB) seem to kick-start the aTreg specification program [24,25,26,27], whereas Basic Leucine Zipper Transcription Factor (BATF), B-lymphocyte-induced maturation protein 1 (Blimp1), JunB, and Myb are then required to establish and maintain the aTreg landscape [24,28]. Importantly, negative regulators of the aTreg program have been described, such as Forkhead Box protein O (Foxo) or E-protein transcription factors [29,30]. At a more advanced level of differentiation, aTreg cells can acquire the expression of tissue- or microenvironment-induced transcription factors historically linked to the polarization of Tconv cells into Thelper (TH) subsets. Indeed, T-Box Expressed in T cells (T-bet), GATA Binding Protein 3 (GATA3), Retinoic Acid Receptor (RAR)-related Orphan Receptor (ROR)γt, and B-Cell Lymphoma 6 protein (Bcl6), hallmarks of TH1, TH2, TH17, and T follicular helper cells, respectively, can also be expressed by aTreg cells and contribute to the diverse and highly adaptive functions of this population [31]. This precise molecular regulation of Treg cell identity has major implications for the regulation of cancer immunity that we will now emphasize.

## 3. Signaling Pathways and Transcription Factors in Treg Cells in Cancer

### 3.1. Foxp3

Because of its critical function in shaping the Treg cell transcriptome and function, Foxp3 itself is an attractive target for cancer immunotherapy. Several groups have identified small-molecule inhibitors able to impair Foxp3 nuclear translocation dimerization or its interaction with known partners such as Nuclear Factor of Activated T-cells (NFAT) [32,33,34]. Interestingly, all these inhibitors had a significant therapeutic effect on syngeneic murine tumors when administered with a peptide vaccine or anti-Programed cell Death 1 (PD-1) checkpoint-blockade therapy. However, one of the major obstacles to further developing such therapies is the expression of FOXP3 by non-Treg cells in humans [35,36]. In recent years, alternative strategies have aimed at targeting Foxp3 cofactors or other pathways, transcription factors, and epigenetic regulators involved in Treg cell biology in cancer; these will be the focus of this review.

### 3.2. IRF4

*Irf4* (encoding IRF4), whose expression in T cells is driven by TCR engagement, is important in many aspects of CD4^+^ and CD8^+^ T-cell biology [37]. In addition, IRF4 is one of the master regulators of aTreg generation [24] and was described as critical for the maintenance of peripheral immune homeostasis. Indeed, conditional ablation of *Irf4* in Treg cells leads to a multifocal and lethal autoimmune disease [38]. More recently, it was shown that IRF4 expression also defines the activated subset of tumor-infiltrating Treg cells in patients with lung cancer and may even regulate the human aTreg phenotype [39]. Furthermore, this tumor-promoting function was demonstrated in mice with tamoxifen-induced ablation of *Irf4* in Treg cells, which display reduced growth of transplanted colon adenocarcinoma [39]. Because IRF4 may also promote T conv cell exhaustion [40], it is emerging as a promising target in cancer.

### 3.3. The Complex Functions of the PI3K/AKT/Foxo Axis

Induced in T cells by CD28 engagement among other signals, the Phosphoinositide 3-Kinase (PI3K)/AKT signaling cascade is well recognized for its critical role in the activation of the Mammalian Target of Rapamycin (mTOR) complex, which serves as a central metabolic rheostat. Aside from this function, AKT also phosphorylates the Foxo1 transcription factor, promoting its exclusion from the nucleus. The function of this axis in Treg cells, especially in the context of cancer, has been under scrutiny in recent years. Historically, it was shown that Treg cells have impaired AKT and mTOR activity, suggesting that the pathway could be expendable or even deleterious for their homeostasis. In support of this latter hypothesis, genetic disruption or chemical inhibition of the phosphatase Phosphatase and Tensin Homolog (PTEN) that leads to exacerbated activation of AKT, drives the conversion of Treg cells into Tconv cells and a significant arrest in tumor growth [41]. Similar results were found upon Treg-restricted ablation of the Neuropilin-1 surface receptor, which normally triggers PTEN recruitment [42,43]. However, at odds with this conclusion, ablation of the PI3K *p110δ* isoform in Tregs, also leads to major delays in the growth of several types of cancers [44,45]. Administration of the p110δ-selective inhibitor PI-3065, as well as other PI3K and AKT inhibitors, has a marked therapeutic effect in preclinical models [44,46]. Similarly, another p110δ inhibitor was shown to strongly inhibit human Treg cell proliferation and function in in vivo assays [47].

Together, this suggests that a balanced activation of the PI3K/AKT pathway is mandatory for the proper suppression of anti-tumor responses by Treg cells. In line with this, both the inactivation and constitutive activation of Foxo1 in Treg cells (using *Foxo1*^flox^ and non-degradable *Foxo1*^CA/CA^ constructs, respectively), lead to multifocal autoimmune syndromes [30,48]. However, *Foxo1*^CA/+^ mice expressing only one copy of the transgene, do not develop autoimmunity but mount strong anti-tumor immune responses, resulting in reduced tumor growth in spontaneous and transplanted cancer models [30]. This could be attributed to a reduced accumulation of Treg cells in the tumor microenvironment. Specifically, Foxo1, while important for the initiation of *Foxp3* expression, is actually an inhibitor of the aTreg cell program. Consequently, Foxo1^CA^ Treg cells fail to upregulate aTreg genes and to migrate to the tumor tissue. In conclusion, even though growing evidence highlights the PI3K/AKT/Foxo axis as an attractive therapeutic target for cancer immunotherapy, one should be cautious when it comes to the specificity, the duration, and the dose–effect of this type of treatment.

### 3.4. Helios

Helios, encoded by *Ikzf2*, is a member of the Ikaros family of transcription factors and expressed by Treg but not Tconv cells in mice [49]. It was originally described that the expression of Helios could be enriched in mouse tTreg when compared to pTreg, though this finding was later debated [50,51]. In addition to this phenotypic hallmark, Helios also supports the identity of Treg cells. Germline deletion or conditional ablation of *Ikzf2* in Treg cells leads to a multifocal autoimmune syndrome detectable from 5 to 6 months of age. Indeed, Treg cells from these animals are present in normal numbers but exhibit impaired suppressive functions in vivo [49,52]. Consequently, melanoma and colon adenocarcinoma growth is significantly delayed in mutant mice [53]. This is associated with an unstable phenotype consisting of enhanced expression of inflammatory cytokines, such as Interferon (IFN)γ or Granzyme B, by tumor-infiltrating Helios-deficient Treg cells [54]. It was also proposed that the expression of Helios by human Treg cells may promote leukemic cell survival and angiogenesis in in vitro assays [55]. Overall, and because it is mainly expendable for Tconv cell function [56], Helios represents an attractive target for cancer immunotherapy. Consistently, it was proposed that agonistic anti-Glucocorticoid-Induced TNFR-Related protein (GITR) antibodies may exert their anti-tumor functions by down-modulating Helios expression in Treg cells [53].

### 3.5. Eos

Eos (encoded by *Ikzf4*), another member of the Ikaros family, is also critical in Treg cell biology. Eos is highly expressed by Treg cells, although in vitro stimulated Tconv cells can express substantial levels of the protein, and physically interacts with Foxp3 to help mediate Foxp3-mediated gene repression in Treg cells [57,58]. Eos is mainly expendable for Tconv cell function in vivo [58]. In contrast, *Ikzf4* knock-down or knock-out drives loss of numerous Treg-specific genes and expression of Tconv-associated genes [57]. Consequently, mice with conditional ablation of *Ikzf4* in Treg cells develop a lymphoproliferative and autoimmune syndrome starting 3 months after birth [59]. This loss of in vivo functionality also leads to impaired suppression of anti-tumor immunity, as shown by the increased T-cell infiltration and function and reduced burden of colon cancer in mutant animals [59].

### 3.6. The Nr4a Family

The Nuclear Receptor Subfamily 4 Group A (Nr4a) family of orphan nuclear receptors is composed of 3 members encoded by *Nr4a1* (Nur77), *Nr4a2* (Nurr1), and *Nr4a3* (Nor1). In T cells, Nr4a expression is induced by TCR signaling in a dose-dependent manner. Interestingly, the expression of the 3 proteins is higher in Treg than in Tconv cells, likely because of the supposedly high autoreactivity of Treg cells [60]. Mechanistically, Nr4a proteins largely regulate the expression of *Foxp3* itself and of Treg-signature genes [61,62]. Nr4a-deficient Treg cells express many inflammatory genes of the TH1 and TH2 (but not TH17) signatures, in particular because Nr4a directly represses the *Il4* locus [63]. Together, these alterations lead to a loss of Treg suppressive function in in vitro assays. Moreover, conditional ablation of the 3 receptors in total T cells almost leads to the complete absence of Treg cells and drives a Th2-mediated lethal autoimmune syndrome early after birth [62]. A similar phenotype was observed in mice with Treg-restricted ablation of all 3 receptors, while ablation of *Nr4a1* and *2* only, was not sufficient to induce disease [63]. These unstable Treg cells are also poor suppressors of anti-tumor immunity. Transplanted tumor growth is significantly reduced in *Nr4a1/2* double mutant animals, in a CD8-T-cell dependent fashion [64]. Importantly, single deletion of *Nr4a1* or *Nr4a2* is not sufficient to induce the phenotype, reinforcing the concept of major redundancy between Nr4a proteins. In this line, an in vitro screening assay revealed that the chemotherapeutic agent camptothecin, a topoisomerase inhibitor widely used in cancer care [65], is a potent inhibitor of Nr4a-induced transcription [64]. Its administration, either alone or with a Cyclooxygenase-2 (COX2) inhibitor, reduces tumor progression in syngeneic transplanted cancers, associated with reduced expression of Nr4a-dependent, Treg-associated genes and enhanced Tconv cell function. Finally, in addition to this effect on Treg cells, inhibition of Nr4a proteins also seems to salvage the dysfunctional state of exhausted anti-tumor T cells, bestowing cells with potent anti-tumor properties [66,67]. Thus, by affecting both Treg cells and exhausted Tconv cells, Nr4a proteins seem to be attractive targets in cancer therapy.

### 3.7. Bach2

Broad Complex-tramtrack-bric a brac and Cap’n’collar Homology 2 (Bach2) is a transcription factor of the Basic Leucine Zipper (bZip) family. Bach2 competes with Activator Protein-1 (AP-1) family members such as Jun and Fos and promotes the repression of AP-1 target genes, thereby restraining the terminal differentiation of CD8^+^ T cells and reducing *Il2* expression in CD4^+^ T cells [68,69]. Bach2 is also critical for the development and homeostasis of Treg cells. Its germline deletion drives systemic lethal autoimmunity in adult mice, associated with reduced Treg cells in the periphery [70,71]. Mechanistically, Bach2 was shown to prevent excessive activation of the rTreg pool by limiting the sensitivity to TCR signals, repressing aTreg-associated genes, and limiting cell apoptosis. For instance, Bach2 restricts the transcription of IRF4 target genes such as *Positive Regulatory Domain Containing 1, with zinc finger domain (Prdm1)* (encoding Blimp-1), allowing the long-term maintenance of rTreg cells and enabling their suppressive activity [72]. Bach2 expression is stabilized at the protein level by SUMO-specific Protease 3 (SENP3), which directly deSUMOylates Bach2 and prevents its nuclear export [73]. This central function has major implications in cancer immunity, since tumor growth is significantly delayed in *Bach2*-null mice transplanted with melanoma or thymoma cells, which is associated with reduced Treg cell accumulation in the tumor [74]. Similarly, conditional *Bach2* or *Senp3* ablation in Treg cells leads to enhanced anti-tumor immunity and delayed progression of transplanted colon cancer and melanoma, respectively, demonstrating a Treg-cell-intrinsic function of Bach2 in cancer [73,75].

### 3.8. E-Proteins and Their Inhibitor of DNA Binding (Id) Counterparts

E-proteins are transcriptional activators and repressors, the activity of which is inhibited by Id proteins. The E-proteins E2A and HEB inhibit the transition to the aTreg cell state [29]. Consequently, conditional ablation of *E2A* and *HEB* in mature Treg cells, leads to the accumulation of highly suppressive aTreg cells in the periphery [29]. Conversely, Treg-restricted deletion of *Id2* and *Id3* drives lethal autoimmunity in mice [76]. Paradoxically, forced expression of *Id2* in Treg cells enables mice to mount strong anti-tumor immune responses, with reduced melanoma growth and Treg cell numbers in the tumor bed [77]. This puzzling observation underlines the need for further analyses on the maintenance of the Treg pools through a balance between E-proteins and Id proteins.

### 3.9. STAT3

As the downstream signaling module of many cytokines and soluble factors, Signal Transducer and Activator of Transcription-3 (STAT3) appears to be an important regulator of Treg cell accumulation and function in cancer. Together with STAT5, STAT3 stabilizes *Foxp3* expression in response to IL-2 [78]; IL-23-driven STAT3 activation in tumor-infiltrating Treg cells achieves the same phenotype [79]. Furthermore, STAT3 and Foxp3 co-operate to promote the suppression of Th17 immunity by Treg cells, as demonstrated by the TH17-mediated intestinal inflammation observed in mice with conditional ablation of *STAT3* in Treg cells [80]. Inhibition of Janus Kinase 2 (JAK2), which phosphorylates and activates STAT3, or STAT3 itself using a chemical inhibitor or antisense oligonucleotides, respectively, reduces the growth of transplanted tumors in mice, associated with decreased Treg cell frequencies [81,82]. However, it is unclear whether this therapeutic effect actually relies on Treg cell inhibition in these settings. Indeed STAT3 is involved in the differentiation and function of many other cell types, including TH17 cells that can be protective or deleterious in cancer, depending on the tumor type (reviewed in [83]). Therefore, additional studies are required to elucidate the cell-autonomous functions of STAT3 in Treg cells and other immune cells during cancer.

### 3.10. The NF-κB Pathway

In the last few years, the NF-κB pathway has emerged as a nexus regulator of Treg cell differentiation, gene expression, and function in autoimmunity and cancer. NF-κB is a family of transcription factors composed of 5 subunits that associate as homo- or heterodimers and all share a Rel Homology Domain (RHD): NF-κB1 (p105/p50, encoded by *Nfkb1*), RelA (encoded by *Rela*), and c-Rel (encoded by *Rel*) of the canonical pathway and NF-κB2 (p100/p52, encoded by *Nfkb2*) and RelB (encoded by *Relb*) of the alternative pathway [84]. In T cells, the canonical pathway is classically activated by the engagement of the TCR and its CD28 partner. Briefly, TCR/CD28 triggering leads to the activation of Protein Kinase C, Isoform Theta (PKCθ). Activated PKCθ phosphorylates Caspase Recruitment domain-containing Membrane-Associated guanylate kinase protein-1 (Carma1), enabling the formation of the CBM complex composed of Carma1, B-cell lymphoma/leukemia 10 (Bcl10), and Mucosa-Associated Lymphoid Tissue lymphoma translocation protein 1 (Malt1). The CBM complex can then recruit the IκB kinase (IKK) complex (composed of IKKα, β, and γ), which phosphorylates the Inhibitor of NF-κB (IκB) proteins to relieve inhibition of NF-κB1, RelA, and c-Rel subunits, allowing their nuclear translocation. Both the canonical and the alternative pathway can also be activated by several members of the TNFRSF and CD28 families, in an activation mechanism that involves the TNFR-Associated Factor (TRAF) family of adaptors.

#### 3.10.1. The Many Roles of NF-κB in Treg Cells

The alternative NF-κB pathway exhibits limited functions in Treg cell homeostasis. Conditional ablation of *Nfkb2*, but not *Relb*, in Treg cells, leads to a mild lymphoproliferation and colon inflammation [85]. *Nfkb2*-deficient Treg cells display a slight reduction in their suppressive function and are mostly characterized by an increased expression of TH17-associated genes, including *Il17a*. In contrast, the canonical pathway is involved in many different aspects of Treg cell biology. The use of Treg-restricted conditional ablation of the canonical pathway intermediates and subunits has revealed the fundamental roles of NF-κB in mature Treg cells in the periphery. Deletion of either *Carma1*, *Bcl10*, *Malt1*, *Ikbkb* (IKKβ), or *Rela* leads to a lethal autoimmune syndrome [86,87,88,89,90]. Interestingly, *Rel* mutant mice display a milder, non-lethal inflammatory syndrome that only arises from >20 weeks of age [90]. This indicates a selective and differential contribution of each NF-κB subunit in the maintenance of peripheral Treg cell homeostasis.

Importantly, these critical functions have a major impact in the context of cancer. Transplanted melanoma growth is dramatically impaired in mice carrying a germline ablation or a Treg-restricted ablation of *Rel*, but not *Rela* [26]. Similar results were obtained when using conditional ablation of *Carma1*, *Bcl10*, and *Malt1* and transplanted models of melanoma, thymoma, or colon adenocarcinoma [86,88,89]. Mechanistically, CBM- and *Rel*-deficient Treg cells exhibit impaired expression of aTreg-associated genes and accumulation in tumor tissues, while significantly increasing their expression of Tconv-associated markers, in particular IFNγ. This even leads to a dominant, Treg-cell-intrinsic anti-tumor function, as the sole transfer of *Carma1*-deficient Treg cells into immunocompetent recipients, is sufficient to reduce tumor growth in an IFNγ-dependent manner [86]. Of note, the transcriptional program of c-Rel- and Carma1-deficient Treg cells does not fully overlap. Moreover, the expression of an IKKβ^CA^ transgene, that leads to the constitutive activation of p50 and RelA (but not c-Rel, personal communication) fails to rescue the autoimmune syndrome in *Carma1* and *Bcl10* mutant animas [86,89]. It is therefore possible that the function of the CBM complex, in this setting, may partially rely on NF-κB-independent functions.

These observations have led to the use of CBM, IKKβ, and c-Rel inhibitors as therapeutic agents in preclinical models of cancer, with encouraging observations. Administration of the Malt1 inhibitors mepazine or MI-2 significantly delays tumor growth when injected in a curative schedule, alone or in combination with a tumor vaccine. The drug also sensitizes checkpoint-blockade-resistant tumors to anti-PD-1 therapy [86,89]. Interestingly, mepazine was proposed to exert its anti-cancer properties through the inhibition of c-Rel in human pancreatic cancer xenografts [91]. Consistently, prolonged systemic inhibition of IKKβ by the Kinase Inhibitor of NF-κB-1 (KINK-1) agent, though ineffective when administered alone, potentiates the anti-cancer effect of a tumor vaccine in transplanted melanoma [87]. Another well-known, though poorly selective, IKK inhibitor, curcumin, decreases the number of intra-tumoral Treg cells in preclinical models of cancer [92,93,94], and enhances Th1 cells, while decreasing Treg cells in patients with lung and colon cancer [95,96]. Finally, c-Rel inhibition using the methylxanthine derivative Pentoxifylline (PTXF) or the selective c-Rel inhibitors, IT-603 or R96A, significantly reduces tumor growth and synergizes with anti-PD-1/Programed cell Death Ligand 1 (PD-L1) therapy as well as doxorubicin [26,97,98,99]. Interestingly, it seems that most of these drugs, in particular mepazine and PTXF, act through the destabilization of tumor-infiltrating Treg cells by inducing their expression of effector cytokines, rather than by depleting the cells or directly perturbing their inhibitory program.

#### 3.10.2. NF-κB Is a Nexus Regulator of the Treg Cell Transcriptional Program in Cancer

In addition to these observations, it is interesting to note that not only does NF-κB directly control many Treg-hallmark genes [90], but it also drives the expression of other transcription factors and epigenetic regulators involved in the suppression of cancer immunity by Treg cells (Figure 2). For instance, NF-κB controls the expression of *Ikzf2* and *Ikzf4* in a RelA- and c-Rel-dependent manner. *Enhancer of Zeste Homolog 2* (*Ezh2*), whose function will be detailed below, is a c-Rel-dependent gene in T cells [100]. Similarly, expression of *Bach2* and *Irf4* is lost in c-Rel-deficient B cells [101,102], though similar observations are currently lacking for T cells.

Collectively, canonical NF-κB and its upstream canonical pathway are of utmost interest for cancer therapy. Of course, owing to the crucial role of NF-κB in cancer initiation, progression, and dissemination [103], many attempts have been made to introduce NF-κB inhibitors into clinics [104]. However, the numerous adverse effects of these inhibitors originally tested in patients have eventually discouraged this research [105]. For further developments in this field, it is thus important to keep in mind that the specificity and safety of such drugs is mandatory for their clinical application. In light of our recent reports demonstrating a selective contribution of each NF-κB subunit to different aspects of Treg cell biology, an attractive option may be to specifically inhibit c-Rel instead of targeting general upstream regulators. No severe adverse effects of the c-Rel inhibitor PTXF (commercialized as Trental) were reported when administered to humans [106]. The recent description of potent and apparently safe c-Rel inhibitors further advocates for such a use [97,99].

## 4. Epigenetic Programing of Treg Cells in Cancer

In addition to the regulation of Treg cell biology by transcription factors, their identity is also linked to the activity of a number of epigenetic regulators, with implications for their targeting in cancer.

Among these regulators, the histone methyltransferase EZH2, is a prime determinant of Treg activity. As part of the Polycomb Repressive 2 (PRC2) complex, EZH2 represses gene transcription through the addition of H3K27^3me^ marks. Importantly, in T cells, *Ezh2* expression is enhanced by CD28 signaling, in a c-Rel-dependent manner [100]. Foxp3 and EZH2 have been shown to co-localize on Treg-associated gene loci, providing a possible mechanism for the repressive activity of Foxp3 [107]. Conditional ablation of *Ezh2* in Foxp3^+^ cells drives a spontaneous lethal inflammatory syndrome linked to Treg cell instability. Moreover, *Ezh2*-deficient naïve T cells display an impaired ability to form Foxp3^+^ iTreg cells in vitro [108,109]. In addition, these unstable Treg cells are poor suppressors of anti-tumor responses, as conditional knock-out (KO) animals show impaired growth of transplanted melanoma colon, prostate, and bladder tumors, associated with enhanced T-cell infiltration and function in the tumor microenvironment [110,111]. Conversely, Treg-restricted ablation of the H3K27 demethylase Jumanji domain-containing protein D3 (*Jmjd3*), increases tumor growth [110]. This set of data is of particular interest, as the efficacy of small-molecule inhibitors of EZH2 for the treatment of cancer is currently under investigation [112]. In support of such strategies, EZH2 inhibitors reduce tumor progression and synergize with anti-CTLA4 therapy in a T-cell-dependent fashion [111,113]. Another layer of epigenetic regulation is the critical role of Histone Deacetylases (HDACs) in Treg cell identity. Four different classes of HDACs are known to repress gene expression and protein stability in various ways and can therefore be either beneficial or deleterious for Treg cells [114]. The class I HDAC inhibitor Entinostat, currently under investigation in different types of cancer, was shown to decrease *Foxp3* expression in vitro and in vivo [115]. Similarly, chemical inhibition or genetic ablation of REST Corepressor 1 (*Rcor 1*) in Treg cells that leads to disruption of HDAC1 and HDAC2 activity, promotes the expression of inflammatory cytokines by Treg cells, and impairs their ability to suppress anti-tumor immunity [116]. Finally, it was shown that genetic ablation of the histone/protein acetyltransferase *p300* specifically in Treg cells, delays the growth of transplanted tumors, while autoimmune symptoms remain mild and non-lethal [117].

Together, these studies argue in favor of the importance of epigenetic regulation in Treg cells during cancer. It is interesting to note that many of the inhibitors tested in the above studies do not seem to affect anti-tumor Tconv cells, revealing Treg-cell-specific requirements for EZH2 or p300 in Treg genome architecture and gene expression. Because many epigenetic modulators are currently tested in the clinics [118], this provides an additional rationale for their use as Treg-cell-targeting agents in cancer immunotherapy.

## 5. Shared Mechanisms of Treg Cell Reprogramming in Cancer

It can appear surprising that so many individual transcription factors and pathways exhibit non-redundant functions in Treg cells during cancer. One explanation may reside in the following common observation: whereas the number of Treg cells and/or their function remains generally intact or only slightly impacted, their stability and their “identity” is largely impaired. Specifically, the conversion of rTreg into aTreg and/or the stability of the aTreg subset is deeply affected by ablating the CBM/NF-κB axis or upon constitutive activation of Foxo1 [26,30,86]. More interestingly, discrete alterations in transcription factors and upstream signals lead to a common increase in the expression of major inflammatory cytokines. Specifically, *Carma1*, *Rel*, *Nrp1*, *Ezh2*, *Senp3*, and *Ikzf2*-null Treg all produce appreciable amounts of IFNγ and/or TNF in the tumor microenvironment [26,43,53,73,86,110] (Figure 2). This transformation into anti-tumor cells was nicely demonstrated with *Carma1*-deficient Treg cells, the transfer of which was sufficient to delay tumor growth in recipient animals [86]. Hence, this highlights Treg “reprogramming” as a relevant and dominant approach to improve anti-tumor immunity [119].

## 6. Challenges on the Path toward Cancer Therapy

We have now at our disposal an exhaustive list of transcription factors involved in Treg-cell-mediated suppression of anti-tumor immunity. Although this highlights many possible molecular targets, there are still a number of challenges that must be considered if we are to apply these discoveries to cancer therapy.

### 6.1. Missing Pieces of the Puzzle

Despite the extensive and thorough knowledge on Treg cell regulation in cancers described above, this obviously remains incomplete. It is likely that other transcriptional regulators will be described in the near future as important for tumor immunosuppression and as putative therapeutic targets. The specific function in cancer, of many master regulators of Treg cell identity and differentiation, has not yet been investigated. For example, the Interleukin-2 (IL-2)/Signal Transducer and Activator of Transcription 5 (STAT5) axis is critical for Treg cell differentiation and function [120,121], but its role in cancer has not been formally evaluated yet. Downstream of the TCR/CD28 signaling, NFAT is integral for the transcriptional landscape of Treg cells [122]. AP-1 transcription factors, in particular JunB, seem to be involved in the specification of aTreg cells [28]. Similarly, the role of Blimp1, a central hub of aTreg cell identity, remains unknown in cancer immunity [123].

It is important to note that many of the studies described in the above sections, used the Foxp3^cre^ mouse strain as a model to investigate Treg cell autonomous functions of given genes. In recent years, it has been largely demonstrated that expression of Cre recombinase can occur in Foxp3^−^ non-Treg cells. This stochastic activity of the *Foxp3* promoter—and “leaky” expression of the Foxp3^cre^ allele—may lead to recombination of the loxP-flanked regions in other immune cells that are involved in cancer immunity, such as CD4^+^ Tconv, CD8^+^ T cells, and also myeloid cell subsets [124,125,126]. Therefore, it would be of the greatest interest to reevaluate some of the functions that were attributed to the sole activity of transcription factors in Treg cells. Different models can be proposed and are already in use: the Foxp3^cre-ert2^ strain that allows tamoxifen-inducible ablation of floxed genes, which limits the duration of Cre activity [127]; the comparison of results obtained with Foxp3^cre^ and other strains such as CD4^cre^ or germline ablations; or the reconstitution of immunodeficient mice with wild-type and mutant immune cell subsets.

In addition, our knowledge will undoubtedly be improved by the implementation of large-scale screening technologies. For instance, three recent studies have used genome-wide and pooled Clustered Regularly Interspaced Short Palindromic Repeats (CRISPR) screen approaches, to identify unknown elements involved in the maintenance of Foxp3 expression and Treg cell identity both in murine and human cells [128,129,130]. These reports have highlighted a series of novel pathways, such as ubiquitin ligases and deubiquitinases, chromatin remodelers, and transcription factors; in particular, Loo et al. nicely demonstrated that ablation of the *Bromodomain-containing protein 9* (*Brd9*) subunit of the non-canonical BRG1/BRM Associated Factors (BAF) complex, impaired *Foxp3* expression and reduced Treg cell function, leading to enhanced tumor immunity and delayed growth of transplanted MC38 colon adenocarcinoma cells [129]. The quantity of data brought by modern technologies such as CRISPR screens will be invaluable for the discovery of unique or shared molecules that can be targeted to impair Treg cells specifically in tumors.

### 6.2. Avoiding Autoimmunity

Immune-related adverse events, in particular autoimmunity, are among the main complications of the checkpoint-blockade cancer therapies anti-CTLA-4 and anti-PD-1. In patients, CTLA-4 blockade often leads to severe autoimmune symptoms, in particular colitis. Although PD-1/PD-L1 blockade generally induces milder complications, thyroiditis is a frequent complication [131]. Mechanistically, there are commonalities between the two treatments: both the engagement of CTLA-4 and PD-1 drive the recruitment of the phosphatase SHP-2 (Src Homology region 2 domain-containing Phosphatase-2) (and PP2A (Protein Phosphatase 2A) for CTLA-4), leading to impaired activation of the TCR/CD28 cascade through PI3K, MAPK, AP-1, and NF-kB [132,133]. Inhibition of either receptor leads to enhance activation of the aforementioned pathways [133,134,135], which might thus be responsible for the autoimmune symptoms. It is therefore of the utmost interest to understand the role of these signaling cascades both in Treg and Tconv cells.

In addition, these symptoms are likely due to the stimulation of T cells not only at the tumor site but also in peripheral tissues. An important step in the development of safe therapeutics would, therefore, be to identify targets that are restricted to the cancer microenvironment. It is noticeable that in many instances, the same pathway or transcription factor can be involved in Treg cell function both in the settings of cancer and tolerance to self. As described in Table 1, Treg-specific ablation of a single gene can lead to pronounced autoimmune symptoms. In contrast, mice with conditional ablation of *Ikzf2*, *Pten*, *Rcor* 1, or *Rel*, for instance, display no (or limited) signs of autoimmunity while enabling enhanced tumor clearance, perhaps indicating “safer” therapeutic targets in terms of adverse effects.

### 6.3. The Issue of Multiple Cell Type Targeting with a Single Agent

The vast majority of the studies described in Section 3 have used Treg-cell-restricted deletion of gene(s) as models. However, it is obvious that many of these transcription factors and regulators also are expressed by other cell types and may impact their function. To ensure the specificity and potency of novel therapeutics, we should at some point obtain a global view of the molecular regulation of the tumor microenvironment as a whole, i.e., assess the potential role of these pathways in other immune cells, as well as in tumor and stromal cells. In some cases, small-molecule inhibitors may impair the fitness of both Treg cells and tumor cells, thus exhibiting a double beneficial effect to restrict cancer progression. This is clearly illustrated, for the various inhibitors of the NF-κB pathway described above. Indeed, in addition to their effect on Treg cell stability, mepazine, KINK-1, or PTXF have all been shown to directly impair tumor cell proliferation, gene expression, and invasiveness [137,138,139,140,141]. However, it is important to keep in mind that signaling pathways and transcription factors can also be expressed and required by anti-tumor immune cells, such as Tconv cells. Once again, this is the case for NF-κB, well-described as a critical component of Tconv cell function in many settings, including cancer [142,143,144,145]. This need for a global delineation of the different players is further illustrated by the importance of the balance between Treg and Th17 cells in many cancer types [146]. Indeed, it appears that the same pathways and transcription factors (e.g., STAT3, NF-κB, or Notch) can be involved in the differentiation and function of both subsets. Thereby, important work remains to be done to dissect the different cell types and mechanisms that can be triggered or targeted by a single agent. This could lead to the use of Treg-cell-directed compounds, for instance, using antibody–drug conjugates that are currently under development [145].

### 6.4. Mouse and Human Treg Cells: Commonalities and Divergences

Mechanistic understanding of the pathways involved in the suppression of tumor immunity by Treg cells has required the use of mouse models. These have enabled the discovery of a myriad therapeutic targets. Nevertheless, it is now essential to compare the expression of transcription factors in murine and human Treg cells and to investigate whether these proteins exhibit similar or divergent functions. This need is exemplified by the absence of consensus on the phenotype of human Treg cells. For instance, FOXP3 can also be expressed by a subset of effector T cells [35,36]. In addition, even the function of human Treg cells in promoting tumor growth is controversial or at least is dependent on the type of cancer [9].

## 7. Conclusions

Different strategies can be considered to impair Treg cell “fitness” in cancers. Protocols aiming at impairing their numbers (recruitment, expansion, survival, etc.) or their suppressive function (CTLA-4, inhibitory cytokines, etc.) are under extensive investigation. The amount of literature described above now suggests that by modulating the transcriptional profile of Treg cells, their potential instability can be exploited in a therapeutic manner. Indeed, instead of depleting Treg cells, strategies aiming at reprogramming Treg cells into anti-tumor cells deserve further development. It may seem optimistic to find the “perfect” mechanism, i.e., a targetable pathway or transcription factor selectively involved in human Treg (but not Tconv) cells, specifically in the tumor (but not normal tissues). However, novel technologies, such as single-cell sequencing and CRISPR/Cas9 approaches, should help unravel fundamental mechanisms in the biology of human Treg cells in cancer, and thus provide a strong rationale for the use of modulators of the transcriptional process for cancer therapy.

## Figures and Tables

**Figure 1 cancers-12-03194-f001:**
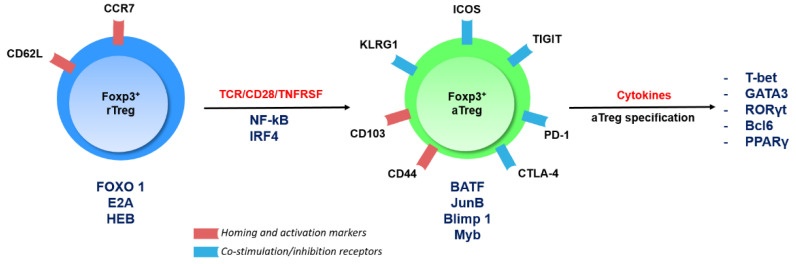
Activation steps of regulatory T cells (Treg cells) and their transcriptional regulation. The figure depicts the main transcription factors involved in the programming and maintenance of Treg cell subsets and highlights the expression of a numbers of surface receptors being targeted in cancer immunotherapy.

**Figure 2 cancers-12-03194-f002:**
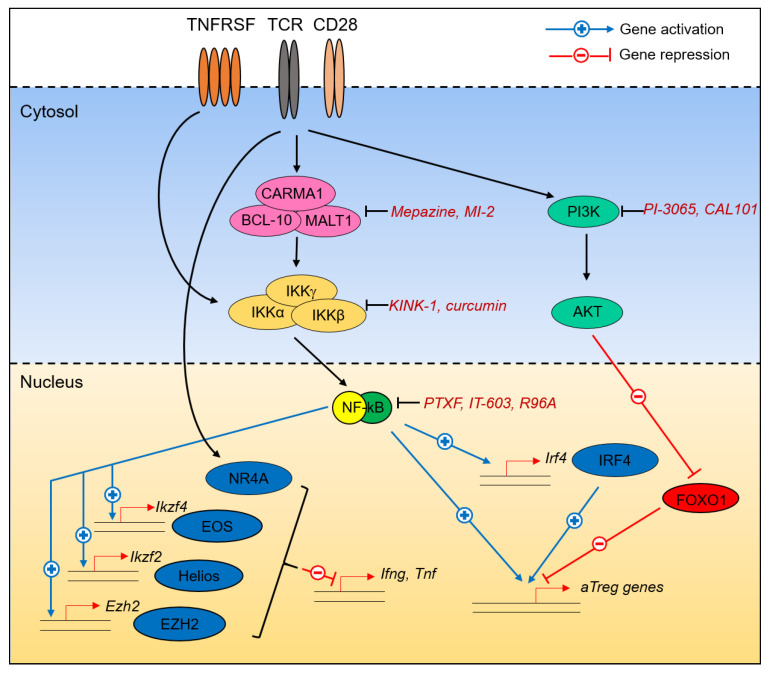
Molecular programming of Treg cells in cancer. Different pathways and transcription factors converge to maintain Treg cell identity in tumors, i.e., the expression of Activated Treg cells (aTreg)-signature genes and the inhibition of inflammatory cytokine expression. Nuclear Factor Kappa-light-chain-enhancer of activated B cells (NF-κB), through its regulation of downstream targets, occupies a central place in this program. A number of chemicals, natural products, and small-molecule inhibitors with potent anti-tumor properties are also highlighted.

**Table 1 cancers-12-03194-t001:** Autoimmune symptoms in mice with enhanced anti-tumor immunity. Models are ranked by severity of the syndrome.

Protein/Family	Mouse Model	Autoimmune Symptoms	Lethality	Ref.
IKKβ	Foxp3^cre^Ikbkb^F/F^	Scurfy-like syndrome	Yes	[87]
Ezh2	Foxp3^cre^Ezh2^F/F^	From 1 month of age, spontaneous T conv cell activation and tissue infiltration	Yes	[108]
IRF4	Foxp3^cre^Irf4^F/F^	From 6 to 8 weeks of age, spontaneous T conv cell activation and tissue infiltration	Yes	[38]
Id proteins	Foxp3^cre^Id2^Tg^	From 6 to 8 weeks of age, progressive T conv cell activation and tissue infiltration	Yes	[77]
STAT3	Foxp3^cre^Stat3^F/F^	From 6 to 8 weeks of age, splenomegaly and colon inflammation	Yes	[80]
Bach2	Bach2^−/−^	From 3 months of age, spontaneous T conv cell activation and tissue infiltration	Yes	[70]
P300	Foxp3^cre^ep300^F/F^	From 10 weeks of age, spontaneous T conv cell activation and tissue infiltration	n.d.	[117]
CBM complex	Foxp3^cre^Bcl10^F/F^	Scurfy-like syndrome	Yes	[89]
	Foxp3^cre^Malt1^F/C4712A^	from 6 to 8 weeks of age, spontaneous T conv cell activation and tissue infiltration	n.d.	[88]
	Foxp3^cre^/^+^Carma1^F/F^	No detectable signs of autoimmunity (Foxp3^cre^/^cre^ Carma1 ^F/F^ develop Scurfy-like syndrome)	No	[86]
NF-κB	Foxp3^cre^Rel^F/F^	Mild lymphoproliferation from 20 weeks of age	No	[90]
Helios	Foxp3^cre^Ikzf2^F/F^	From 5 months of age, spontaneous T conv cell activation and tissue infiltration	No	[52]
Foxo1	Foxp3^cre^Foxo1^CA/+^	No detectable signs of autoimmunity (Foxo1^CA/CA^ develop lethal autoimmunity)	No	[30]
Pten	Foxp3^cre^PTEN^F/F^	From 12 weeks of age, spontaneous T conv cell activation, kidney damage	No	[136]
	Foxp3^cre^PTEN^F/F^ (2nd construct)	No detectable signs of autoimmunity	No	[41]
Nr4a	Foxp3^cre^Nr4a1^F/F^Nr4a2^F/F^ dKO	No detectable signs of autoimmunity (Nr4a1/3 dKO and Nr4a1/2/3 tKO develop lethal autoimmunity)	No	[63]
PI3K	Foxp3^cre^p110d^F/F^	No detectable signs of autoimmunity	No	[44]

n.d.: not described; dKO: double knock-out; tKO: triple knock-out.

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
