# Peer review of "Transcriptional Control of Regulatory T Cells in Cancer: Toward Therapeutic Targeting?"

_cancers, 2020, doi:10.3390/cancers12113194_

Round 1

Reviewer 1 Report

In this review, the authors present a summary and critical analysis of the recent literature regarding the different signaling pathways involved in the programming of T reg cell homeostasis and functions in cancer. They also highlight the therapeutic approaches aiming at targeting these regulators to increase anti-tumoral immunity without inducing auto-immune or inflammatory disorders. To complete this review, we suggest to the author to take into account the recent studies that highlight the weakness Foxp3Cre mice model used by several studies in Treg field. Indeed, it is now well established that the expression of Cre recombinase may not always be restricted to the target cell or tissue of interest due to promiscuous activity of the driving promoter. Thus, the expression of Cre recombinase and, by extension, excision of the loxP-flanked gene may occur in non-target cells. Recently, by breeding the Foxp3 promoter Cre recombinase mouse line to the floxedstopped tdTomato reporter strain, it has been shown that Foxp3-driven Cre recombinase induced tdTomato red fluorescent protein in Treg cells but also in lymphoid (B cells and CD8 T cells) and blood-resident myeloid cells (dendritic cells, monocytes, neutrophils) suggesting stochastic activity of the Foxp3 promoter (https://doi.org/10.3389/fimmu.2019.02228). Therefore, the interpretation of the data using this model in the Treg filed should be revaluated.

To facilitate the reading of the review, the author should distinguish clearly between the impact of the described factors on Treg development in the thymus and Treg plasticity/stability.

Author Response

In this review, the authors present a summary and critical analysis of the recent literature regarding the different signaling pathways involved in the programming of T reg cell homeostasis and functions in cancer. They also highlight the therapeutic approaches aiming at targeting these regulators to increase anti-tumoral immunity without inducing auto-immune or inflammatory disorders.

We thank the reviewer for his/her positive evaluation of our manuscript, and have assessed his/her comments below and in the text.

  • To complete this review, we suggest to the author to take into account the recent studies that highlight the weakness Foxp3Cre mice model used by several studies in Treg field. Indeed, it is now well established that the expression of Cre recombinase may not always be restricted to the target cell or tissue of interest due to promiscuous activity of the driving promoter. Thus, the expression of Cre recombinase and, by extension, excision of the loxP-flanked gene may occur in non-target cells. Recently, by breeding the Foxp3 promoter Cre recombinase mouse line to the floxedstopped tdTomato reporter strain, it has been shown that Foxp3-driven Cre recombinase induced tdTomato red fluorescent protein in Treg cells but also in lymphoid (B cells and CD8 T cells) and blood-resident myeloid cells (dendritic cells, monocytes, neutrophils) suggesting stochastic activity of the Foxp3 promoter (https://doi.org/10.3389/fimmu.2019.02228). Therefore, the interpretation of the data using this model in the Treg filed should be revaluated.

We thank the reviewer for this highly relevant observation. We have added a section in paragraph 6.1. (line 405) to acknowledge the need for alternative models in order to fully delineate the Treg-cell autonomous roles of pathways and transcription factors.

  • To facilitate the reading of the review, the author should distinguish clearly between the impact of the described factors on Treg development in the thymus and Treg plasticity/stability.

We thank the reviewer and agree with his/her comment. To avoid confusion, we have deleted the sentences describing putative roles of TFs in Treg cell development (and the associated references)(lines 188, 214, 229, 275). We are now solely focusing the review on the regulation of mature Foxp3+ Treg cells, as stated in the Introduction (line 65).

Reviewer 2 Report

In this manuscript, Stéphan et al. make a clear and interesting review of the transcriptional control of regulatory T cell development and function with special emphasis on cancer therapy. The manuscript is very well organized and of a very good quality. References are appropriate and updated. English spelling need to be reviewed. In conclusion, I consider that the work can be accepted for publication in Cancers.

Author Response

We thank the reviewer for the positive evaluation of our paper. 

Reviewer 3 Report

In this paper, the authors discuss about understanding of significances of transcriptional modulation of Treg cells in cancer therapy. It is a well-written and organized review article nicely conveying current insights into the present status and future direction of anti-tumor approach. I suggest the authors to add brief descriptions more about the following aspects, which is hopefully able to further raise the strength of present manuscript.

  • More discuss about recent methodological breakthrough to screen critical factors modulating FoxP3 in Treg cells (using CRISPR) by citing relevant reference(s).
  • Also briefly mention about transcriptional control of expression of regulatory cytokines such as TGF-beta or IL-10 in Treg cells if there are any reports
  • Describe more about impact of the balance of Treg and Th17 on contributing to tumor prognosis or progression (Notch, STAT3 engaged?), although 3.9.1 section bears a slight description
  • Discuss more about any possible side effects of treating inhibitors of immune checkpoints (CTLA-1, PD-1, PD-L1) in cancer therapeutics

Minor points

  • In vitroin vitro (i, small letter) on line 60
  • Make a space between “transcription” and “[60]” on line 194
  • Delete a space between “cell” and “s” and make a space between “s” and “[93]” on line 296
  • Check the word “Repressor” (isn’t it “Repressive”??) on line 320

Author Response

In this paper, the authors discuss about understanding of significances of transcriptional modulation of Treg cells in cancer therapy. It is a well-written and organized review article nicely conveying current insights into the present status and future direction of anti-tumor approach. I suggest the authors to add brief descriptions more about the following aspects, which is hopefully able to further raise the strength of present manuscript.

We thank the reviewer for his/her positive evaluation and helpful comments; we have largely edited the manuscript to address the reviewer’s comments.

  • More discuss about recent methodological breakthrough to screen critical factors modulating FoxP3 in Treg cells (using CRISPR) by citing relevant reference(s).

We thank the reviewer for his/her excellent suggestion and have included a description of the recent CRISPR-screen approaches used for unraveling the complexity of Treg cell stability, in section 6.1 (line 417).

  • Also briefly mention about transcriptional control of expression of regulatory cytokines such as TGF-beta or IL-10 in Treg cells if there are any reports

The reviewer raises a good point here, as both TGF-b and IL-10 are critical for Treg cell function. However, to date and to our knowledge, there is no clear consensus on the molecular mechanism driving specific expression of these cytokines in Treg cells. TGFb mRNA and protein expression is a multifactor mechanism that may involve the MAPK and/or the the PI3K/AKT pathway; whereas Il10 expression may require ERK and/or SMADs and/or Jun and/or NF-kB depending on the context, as emphasized by the numerous transcription factor binding sites found in the Il10 locus. Because of this complexity, we have chosen not to discuss the regulation of these cytokines in our report.

  • Describe more about impact of the balance of Treg and Th17 on contributing to tumor prognosis or progression (Notch, STAT3 engaged?), although 3.9.1 section bears a slight description.

We thank the reviewer for his/her suggestion. The balance between Th17 and Treg cells and its molecular control is indeed of interest. In consequence, we have added a new section on the role of STAT3 in Treg cells during cancer (new section 3.9, line 238; and Table 1), line, as well as a couple sentences of open discussion in section 6.3 (line 470).

  • Discuss more about any possible side effects of treating inhibitors of immune checkpoints (CTLA-1, PD-1, PD-L1) in cancer therapeutics 

We thank the reviewer for his/her suggestion. We have further discussed Treg-related adverse effects of CTLA-4 and PD-1 targeting agents, and their molecular mechanisms, in the introduction (line 54) and in section 6.2 (line 435).  

Minor points

  • In vitro→ in vitro (i, small letter) on line 60

We have edited the text accordingly.

  • Make a space between “transcription” and “[60]” on line 194

We have edited the text accordingly.

  • Delete a space between “cell” and “s” and make a space between “s” and “[93]” on line 296

We have edited the text accordingly.

  • Check the word “Repressor” (isn’t it “Repressive”??) on line 320

We have edited the text accordingly.